# An Emerging Role for Epigenetics in Cerebral Palsy

**DOI:** 10.3390/jpm11111187

**Published:** 2021-11-12

**Authors:** Brigette Romero, Karyn G. Robinson, Mona Batish, Robert E. Akins

**Affiliations:** 1Department of Medical and Molecular Sciences, University of Delaware, Newark, DE 19711, USA; bmromero@udel.edu; 2Center for Pediatric Clinical Research and Development, Nemours Children’s Health System, Wilmington, DE 19803, USA; karyn.robinson@nemours.org

**Keywords:** epigenomics, DNA methylation, histone modification, noncoding RNA, spasticity, neonatal encephalopathy

## Abstract

Cerebral palsy is a set of common, severe, motor disabilities categorized by a static, nondegenerative encephalopathy arising in the developing brain and associated with deficits in movement, posture, and activity. Spastic CP, which is the most common type, involves high muscle tone and is associated with altered muscle function including poor muscle growth and contracture, increased extracellular matrix deposition, microanatomic disruption, musculoskeletal deformities, weakness, and difficult movement control. These muscle-related manifestations of CP are major causes of progressive debilitation and frequently require intensive surgical and therapeutic intervention to control. Current clinical approaches involve sophisticated consideration of biomechanics, radiologic assessments, and movement analyses, but outcomes remain difficult to predict. There is a need for more precise and personalized approaches involving omics technologies, data science, and advanced analytics. An improved understanding of muscle involvement in spastic CP is needed. Unfortunately, the fundamental mechanisms and molecular pathways contributing to altered muscle function in spastic CP are only partially understood. In this review, we outline evidence supporting the emerging hypothesis that epigenetic phenomena play significant roles in musculoskeletal manifestations of CP.

## 1. Introduction

The goal of most personalized medicine efforts is to provide the right treatment at the right time for every patient. This goal has been elusive for clinicians caring for children with cerebral palsy (CP) due in large part to a lack of understanding of fundamental pathways contributing to disease. CP is a set of movement disorders and most individuals with CP are born with it, often without a clear etiology. Recent work has illuminated genomic contributions to CP with estimates ranging from 2 to 30% of cases being linked to heritability or alterations in primary DNA sequence [1,2,3,4]. Complex studies to relate specific genetic disruptions to subtypes of CP are ongoing, but for the majority of cases, the potential molecular etiology of CP has not been revealed. Importantly, while genetics delimits the potential for gene expression by providing requisite DNA sequences, epigenetic pathways control expression itself by allowing or disallowing access to specific DNA regions or by interfering with expression processes. Accumulating data suggest that epigenetic pathways, which alter the expression of genes without changing primary DNA sequence, may also play roles in CP, especially in peripheral manifestations of the movement disorder.

CP is the most common motor disability of childhood [5] with prevalence estimates ranging from about 1 to more than 3 per 1000 children [6,7,8,9,10]. CP results from maldevelopment or injury to the developing brain with associated impairments of movement and posture [11]. The central nervous system lesion in CP is nonprogressive, but neuromuscular function often deteriorates over time in individuals with CP [12]. Individuals with CP present with a spectrum of severities [5] and a set of different subtypes: spastic, dyskinetic/athetoid, ataxic, hypotonic, and mixed. Of these, spastic CP is the most common type, accounting for approximately 80% of all cases [13]. The primary symptoms in spastic CP involve increased muscle stiffness and tone (spasticity), contractures, muscle weakness, and disrupted gait and body positioning [11,14]. These motor impairments are frequently accompanied by additional deficits such as intellectual disability, epilepsy, and sensory impairments [15].

Alterations in muscle tissue have been observed in children with CP. These include impaired longitudinal growth contributing to contractures [16], decreased muscle cross-sectional area [17], increased sarcomere length [13], reduced muscle mass and volume, fibrotic tissue accumulation, increased extracellular matrix, reduced satellite-cell numbers [14], microanatomic disruption of neuromuscular junctions (NMJ) [18], and muscle-type-specific alteration of fiber type distributions [19,20]. In most cases, the specific causes and mechanisms contributing to these alterations have remained elusive.

In this review, we evaluate the state of epigenetic research in CP and summarize recent studies with a particular focus on studies of muscle and neuromotor development. Searches of the PubMed repository (pubmed.ncbi.nlm.nih.gov; (accessed on 27 September 2021)) were conducted for articles relating to “cerebral palsy”, “DNA methylation”, “histone modification”, “miRNA”, “circRNA”, “lncRNA”, “noncoding RNA” with critical features of CP muscle. Identified peer-reviewed, primary journal research articles and reviews published since 2010 are tabulated and discussed in context.

### 1.1. Cerebral Palsy Risk Factors and Relationship to Epigenetics

Several risk factors and potential pre- and peri-natal causes for CP have been identified, including prematurity, hypoxia–ischemia, placental insufficiency, chorioamnionitis or other prenatal infection, perinatal inflammation, genetic causes, and combinations of these factors [21,22,23,24]. Importantly, the types of physiologic perturbation associated with CP are also associated with modification of epigenetic patterning and pathways in multiple tissues and in multiple contexts during fetal and early life [25,26,27,28]. For example, hypoxia during fetal development has been found to change gene expression and to alter epigenomic mechanisms that determine the physiological condition of the fetus [29]. These effects in turn are associated with future responses and with long-term risks for deleterious outcomes [29]. Particularly, perinatal hypoxia has been linked to vascular dysfunction and disease later in life [30,31,32,33].

Principal epigenetic mechanisms that are likely to play important physiologic and developmental roles within the perinatal timeframe associated with the onset of CP include DNA methylation, histone modifications, and noncoding RNAs. The manifestation of CP symptoms, and the long-term health outlook for individuals with CP, could be affected by disruption of the specific epigenomic mechanisms in the fetus. Research in this area is only beginning to take shape, but critical aspects of the complex molecular and epigenetic landscapes impacting the health and disease status of individuals with CP are coming to light. Thus, there is a need to better understand relationships between epigenetic mechanisms and the onset and progression of CP.

### 1.2. Cerebral Palsy Diagnosis

Although many risk factors of CP have been identified, the specific cause of the condition in individuals remains unknown in many cases [1], and diagnosis is often not made until late in the second year of life [34]. A recent study of parents of children with CP found that 60% indicated suspicions of CP under 6 months of age, but only 21% were provided a diagnosis of CP when their child was under 6 months of age [5]. Earlier diagnosis is desirable, as a delay in diagnosis can result in the loss of opportunity for early intervention. Due to the neuroplasticity of the developing brain, early intervention has the ability to positively impact neurodevelopment and functional ability [34]. Currently, diagnosis is based on history taking, neurological examination, and movement assessment [34] along with magnetic resonance imaging (MRI), which is abnormal in more than 80% of children with CP [35], and genetic and biochemical testing to rule out diseases with similar symptoms [35].

The lack of MRI findings in many children with CP, the concurrent presence of congenital anomalies, and the higher rates of CP in monozygotic versus dizygotic twins have led to the hypothesis that much of the unknown pathophysiology of CP may have a genetic cause [36,37]. Recent findings have implicated copy number variants (CNVs) and single nucleotide variants (SNVs) in some subjects with CP. De novo CNVs have been identified in 7–31% of subjects depending on the study [4,15,38,39,40,41]. An SNV study of 1256 subjects identified pathogenic or likely pathogenic variants in 229 genes, 86 of which were mutated in two or more subjects [1]. The overall prevalence of pathogenic and likely pathogenic variants was 32.7% in a pediatric cohort and 10.5% in an adult cohort [1]. A whole exome sequencing study of 250 parent–offspring trios indicated an enrichment of damaging de novo mutations (DNMs) in CP [2]. Eight genes had multiple damaging DNMs, with two (*TUBA1A* and *CTNNB1*) reaching genome-wide significance. Identical DNMs were only identified in two unrelated CP cases in two genes, *RHOB* and *FBXO31*. Overall, the study found that approximately 12% of CP cases could be attributed to an excess of DNMs, and 2% to recessive variants [2].

In addition to genomic variants, recent evidence suggests there are epigenetic alterations in DNA methylation patterns in subjects with CP [42]. Similarly, histone modifications have been associated with CP, and while not yet studied in CP, dysregulation of noncoding RNAs has been found in multiple pathophysiologic states including tumorigenesis as well as neurological, cardiovascular, and developmental diseases [43]; microRNAs, which are noncoding RNAs, play a role in modulating muscle regeneration and satellite-cell behavior [44]. Transcriptional studies support the notion that gene regulatory pathways are disrupted and demonstrate differential gene expression in skeletal muscle of individuals with CP, including in critical pathways associated with muscle growth, metabolism, and extracellular matrix production [45,46]. The possibility that altered DNA methylation, histone modifications, or noncoding RNAs may contribute the pathophysiology of CP is intriguing. Further studies linking motor dysfunction in CP to epigenetic modifications and noncoding RNA alterations are warranted.

### 1.3. Microanatomic Disruption of Muscle in Cerebral Palsy

Functional microanatomic features of muscle fibers, including striated sarcomeres, endo/perimysium, and neuromuscular junctions (NMJ), appear disrupted in CP (for example, see [47,48]). In particular, the microanatomy of NMJs is exquisitely regulated [49]. The NMJ is the site of communication between motor nerves and muscle fibers, and there is a tightly controlled coordination of NMJ structure and function. Functionally, when an action potential reaches the nerve terminal, voltage-gated calcium channels open and the influx of calcium leads to docking of synaptic vesicles at active zones and the release of the neurotransmitter acetylcholine (ACh) into the synaptic cleft. ACh binds to nicotinic ACh receptors (nAChRs) on the motor endplate at the crests of junctional folds in the sarcolemma. Binding of ACh to the receptors causes a conformational change which allows an influx of sodium, depolarizing the muscle membrane and activating a cascade that results in muscle contraction. Acetylcholinesterase (AChE) within the synaptic cleft catabolizes the released ACh to prevent prolonged contraction. The development, maturation, and maintenance of the NMJ require complex molecular interactions between motor neurons, muscle fibers, and perisynaptic Schwann cells, and the involvement of many proteins whose precise organization is critical. Several studies have demonstrated a disorganization of these NMJ components in pediatric subjects with CP [18]. Subjects with CP require different doses of neuromuscular blocking agents (NMBAs), which act at the NMJ, compared to subjects without CP. In particular, subjects with CP are more sensitive to the depolarizing agent succinylcholine and more resistant to the nondepolarizing agents vecuronium and rocuronium [50,51,52]. When the relative distribution of specific presynaptic, synaptic, and postsynaptic markers (presynaptic: SV2; synaptic: AChE, laminin β2; postsynaptic: nAChR) were quantified by fluorescence microscopy in spinalis muscle acquired from scoliotic subjects with or without CP, NMJs in the CP group were found to be disrupted, with all pairwise comparisons demonstrating significant differences except SV2 compared to nAChR [18]. The dysmorphism that was the most significant, present in the greatest proportion of subjects with CP, and most predictive of a diagnosis of CP was the presence of synaptic AChE outside the limits of synaptic laminin β2 [18]. Electron microscopy provided further evidence of NMJ disorganization, indicating that the NMJs of subjects with CP had a significantly greater distance between postsynaptic folds, significantly deeper postsynaptic folds, and a reduced number of presynaptic mitochondria [18]. Transcriptionally, muscles from subjects with CP have demonstrated differential expression of NMJ genes, including significantly increased expression of *HSPG2*, *NID1*, and *COLQ* and significantly decreased expression of *UTRN*, *CHRN*, and *COL4* in gracilis and semitendinosus by Affymetrix HG-U133A 2.0 chips [46] and significantly increased expression of *DMD* in spinalis by RNA-sequencing [45].

### 1.4. Satellite Cells in Cerebral Palsy

Muscles from individuals with CP have also demonstrated disruptions related to satellite-cell number and function. Satellite cells, stem cells residing between the sarcolemma of muscle fibers and the surrounding basal lamina, are mediators of muscle growth and repair [53]. When activated, they proliferate and either self-renew to maintain the satellite-cell pool or differentiate toward myoblasts and fuse with existing myofibers or other satellite cells to form multinucleated myotubes [14,16]. This process is controlled by sequential expression of transcription factors [54]. The classical marker for satellite cells is PAX7, a transcription factor that is expressed in all satellite cells [54] and stops expression after differentiation [16], but satellite cells can be identified at various stages by different additional molecular markers such as PAX3, NCAM, MYF5, MRF4, and MYOD [14]. A flow cytometry study of hamstring muscles from pediatric subjects with contractures related to CP indicated that 5.3% of cells were satellite cells, compared to 12.8% in typically developing children when using NCAM as a satellite cell marker [16]. An immunohistochemical study of gracilis muscles similarly found a 70% reduction in satellite cells in subjects with CP when cross-sections were stained for nuclei, PAX7, and laminin [55]. Because satellite cells are the source of increased myonuclei associated with muscle growth and sarcomerogenesis [55], this reduction in satellite-cell number is believed to be responsible for CP muscle’s inability to add sarcomeres in series, leading to increased sarcomere lengths and decreased longitudinal growth [16].

In vitro studies of myoblasts derived from satellite cells indicated that when differentiated, satellite-cell-derived myoblasts from CP muscle had an 85% decrease in myotube formation and a 74% decrease in fusion index compared to satellite-cell-derived myoblasts from typically developing children [56]. In this study, Western blotting indicated significant downregulation of markers of myoblast differentiation and myotube maturation, including MYH7, MV, and SERCA1 [56]. Real-time qPCR of 85 genes known to be involved in myogenesis and skeletal muscle remodeling demonstrated significant upregulation of 9 genes and significant downregulation of 42 genes, including *ACTA1*, *ACTA2*, *TNNI3*, *DES*, *TPM1*, *MEF2C*, *MEF2A*, *MYF5*, *MYF6*, *CDK5*, *PAX3*, *CAV1*, *CAV3*, *MAPK14*, *CTNNB1*, *SHH*, *ITGA7*, *ITGB1*, and *ITGAV* [56]. An RNA-seq study of satellite-cell-derived myotubes from subjects with CP demonstrated significant upregulation of 28 genes and significant downregulation of 62 genes, including *ITGA5*, *COL6A1*, *COL6A3*, *LAMA4*, *HGF*, and *PDGFRB*, compared to controls [45]. Another study of the differentiation of satellite-cell-derived myoblasts from gastrocnemius demonstrated enhanced fusion index compared to a reduced fusion index in cells derived from subjects with CP, along with disorganized nuclear spreading [14]. These contrasting results on fusion index highlight the heterogeneity of CP, but provide further evidence that there are fundamental differences in satellite-cell function in subjects with CP.

## 2. DNA Methylation

DNA methylation is a primary mode of epigenetic regulation and inheritance and a fundamental mechanism for gene regulation in development and disease [57]. DNA methylation involves covalent modification of DNA bases, typically the methylation of cytosine to form 5-methyl-cytosine in the context of CG dinucleotides (called CpG sites). It is clear that significant physiologic perturbations during gestation or early in life can alter DNA methylation patterns [58,59,60]. The types of physiologic stress that are associated with the onset of CP such as hypoxia, infection, inflammation, and intrauterine growth restriction are all associated with altered DNA methylation in a number of experimental and physiologic contexts, and early life stress can lead to DNA methylation changes in blood cells and other tissues that are sustained later in life [61,62]. For example, early onset pre-eclampsia is associated with altered methylation in cord blood cells [62], and premature birth is also associated with epigenetic modifications of DNA [63,64]. Acute hypoxic exposure of cultured neurons results in persistent DNA methylation changes [65]; human dendritic cells respond to bacterial infection by remodeling DNA methylation [66]; the acute inflammation associated with myocardial infarction leaves differential DNA methylation patterns in individuals [67]; fetal growth restriction has been linked to methylation of placental genes [68]. In general, significant physiologic stress/trauma in childhood can be associated with genome-wide changes in DNA methylation [69].

A search of the PubMed repository (pubmed.ncbi.nlm.nih.gov; (accessed on 27 September 2021)) for articles relating DNA methylation with CP yielded 11 results, including 2 review articles (Table 1). The earliest report identified from Luo et al. [70] evaluated expression of the *Kank1* gene, the deletion of which has been associated with CP in earlier work [71]. The two review articles covered aspects of DNA methylation changes and CP associated with maternal obesity [72] and preterm birth [73], respectively.

Three studies were identified that used samples from monozygotic twin pairs that were discordant for CP (i.e., one twin had a CP diagnosis; the other did not). Jiao et al. [80] examined sets of twins at ~3 years of age using reduced representation bisulfite sequencing on the genomic scale. They identified 190 differentially methylated genes; 37 hypermethylated and 153 hypomethylated in the CP group and identified differentially methylated genes that may be associated with the occurrence and development of CP in twins. Yuan [79] provided a short report drawing attention to differential methylation, especially promoter regions. Mohandas et al. [78] utilized newborn blood spots of twin pairs who later became discordant for CP using Illumina 450 BeadChip arrays. They identified 33 differentially methylated probes and two differentially methylated DNA regions that were associated with CP. Each of these was limited by the small sample sizes available but, together provide compelling support for the notion that CP is associated with altered DNA methylation profiles.

Two of the articles relating CP and DNA methylation addressed differences in muscle cells and tissue. Interestingly, muscle is suspected to have an “epigenetic memory” function wherein altered DNA methylation patterns are associated with muscle reprogramming after stress, disease, or exercise [82,83]. Von Walden et al. [76] evaluated skeletal muscle samples from 19 CP surgical cases compared to postmortem samples from 10 typically developing controls to assess methylation of rDNA promoter. They found that although muscle ribosome biogenesis was suppressed in CP, the methylation status of the promoter was not significantly different between the cohorts. Interestingly, study participants with more severe impairment did show decreased promoter methylation. Two papers by the same group (Domenighetti et al. [56] and Sibley et al. [74]) both looked at satellite cells isolated based on CD56/Ncam1 expression from study participants with muscle contractures requiring surgical release and compared them to cells from non-CP controls. In the first paper, they evaluated the characteristics of these cells in culture and found that altered methylation of a β1-integrin promoter (identified as integrin-β 1D or ITGB1D) was associated with decreased ITGB1D expression and reduced muscle-specific functions in cells from the CP cohort. Treatment with a demethylating agent reversed the DNA-methylation-associated effects and improved myotube formation. The second paper further showed DNA hypermethylation in CP with gene expression patterns that favored cell proliferation over differentiation. Treatment with a hypomethylating agent reduced DNA methylation to control levels and reduced proliferative activity. These articles and concepts implicate altered DNA methylation as a potential contributor to dysfunction in CP muscle.

Two studies, one from members of our group [42] and one by Bahado-Singh et al. [77], employed advanced computational approaches and elements of artificial intelligence/machine learning to evaluate DNA methylation pattern differences in blood cells. In the Bahado-Singh article, DNA from newborn blood spots was analyzed with Illumina Human Methylation 450 K modules. They found 230 differentially methylated loci in 258 genes were that were >2.0-fold altered. The area under the receiver operating curve was 0.75 for CP detection, and an artificial intelligence/machine learning computations approach showed a 95% sensitivity and 94.4% specificity for prediction of CP based on the derived DNA methylation biomarkers. In the study from our group, blood was collected from much older participants (~15 years old) and analyzed using an analytic and computational pipeline based on methylation-sensitive restriction-enzyme digestion, next-generation DNA sequencing, and identification of ~1.5 million potential methylation sites. Among these, we identified 2809 hypomethylated sites and 3779 hypermethylated sites in the CP cohort. An artificial-intelligence/machine-learning analysis based on linear discriminant analyses identified sets of 40 or fewer CpG sites with near perfect accuracy in simulated testing. Surprisingly, these sets of CpGs also provided strong results (>70% accuracy) identifying samples from children who were about 10 years younger.

In fact, DNA methylation appears to be dynamic during development and neuromotor maturation. For example, a meta-analysis of Illumina Human Methylation 450 K data from 3648 typically developing newborns (i.e., non-CP) by Merid et al. [84] identified 8899 CpGs in cord blood that were related to gestational age at birth (range 27–42 weeks). DNA methylation in a subset of those genes also correlated with gestational age in fetal brain and lung tissue, indicating that the DNA methylation landscape is changing during these phases of development. In evaluations of peripheral blood cells of newborns, Massaro et al. [75] found differentially methylated regions in babies with CP that showed changes in methylation level over the first 14 days of life where the same regions in matched controls remained stable. Thus, CP may be associated with both differences in DNA methylation levels and differences in the modulation of DNA methylation patterns in early life.

Altogether, although the field is new and still developing, existing studies support the idea that DNA methylation patterns differ in different tissues in CP. The results also suggest that aspects of these pattern differences may prove useful as biomarkers and that some may even be associated with the onset or progression of disease. Significantly more research is needed to investigate these possibilities.

## 3. Histone Modifications

Histone modifications are primary drivers of chromatin configuration and epigenetic regulation of gene expression. There are multiple physiologically relevant, reversable, post-translational modifications to histones, including acetylation, methylation, and phosphorylation, which have been well-studied, as well as roles for sumoylation, ubiquitylation, O-linked β-N-acetylglucosamination, ADP ribosylation, deimination, and proline isomerization that are still emerging [85,86,87,88]. Histone modifications are critical for muscle development and function, and modifiers of histone modifications are clinically important. Histone deacetylase (HDAC) inhibitors such as Givinostat have been found to enhance muscle regeneration in dystrophic muscle [89,90] and are the subject of Phase III clinical trials (e.g., NCT02851797 on clinicaltrials.gov). In studies of nonhuman models, histone modifications have been identified as important regulators of NMJ structure and function; for example, recent reviews point out that postsynaptic myonuclei exhibit enrichment of decondensed chromatin and acetylated histones in subsynaptic nuclei [91], and HDACs have been specifically implicated in NMJ stabilization in preclinical model systems like *Drosophila* sp. [92,93]. Histone modifications are also implicated in muscle and satellite-cell function including the activation of satellite cells to proliferate and form myotubes, a process that is dysregulated in disease states [94,95,96].

Unfortunately, only limited work has been carried out thus far to evaluate histone modifications in CP. Of interest, though, is the fact that post-translational histone modifications are sensitive to acute hypoxia and chronic ischemia, which have been shown to induce global histone acetylation differences in rodent placenta [97]. A search of the PubMed repository (pubmed.ncbi.nlm.nih.gov; accessed on 27 September 2021) for articles relating histone modification with CP yielded only three articles; one of these was a review (Table 2). Given the published relationships between risk factors for CP and histone modification as well as the associations between histone modification and muscle, additional research is needed in this area.

## 4. Noncoding RNAs

Several regulatory noncoding RNAs (ncRNAs) have been attributed as key players for epigenetic regulation of gene expression in normal physiological processes [100] as well as in a myriad of diseases [43]. The three most well-studied classes of ncRNAs include microRNA (miRNA or miR), long noncoding RNA (lncRNA), and circular RNAs (circRNA) [101,102]. Small interfering RNAs (siRNAs), small transfer RNAs (tRFs), and piwi-interacting RNAs (piRNAs) among others have also been shown to play regulatory roles [103]. Although cerebral palsy (CP) is the most common childhood disability, the role of ncRNAs in CP is only beginning to be appreciated. Accordingly, we performed a comprehensive review of the literature to identify key ncRNAs that have been reported to play a role in CP. In addition, we identified ncRNAs that may be involved in CP based on their association with brain and muscle development. For example, microRNAs (e.g., miR-1) and lncRNAs (e.g., linc-MD1) are included as they play important roles in Duchenne muscular dystrophy (DMD), which is a lethal neuromuscular disease [45,104].

### 4.1. MicroRNAs in CP

MicroRNAs (miRs) are small noncoding RNAs of about 22 nucleotides that are implicated in developmental processes such as neurogenesis and myogenesis, as well as in many pathological processes [105]. Amongst all noncoding RNAs, miRNAs are the most well studied for their regulatory roles in controlling gene expression by either targeting mRNAs for degradation or blocking their translation. The appreciation of their role as potential therapeutic and diagnostic biomarkers for several neural diseases has come only in recent years [106,107]. Li et al. showed that miR-1 was noticeably increased in a rodent CP model, suggesting that it may play a role in CP pathogenesis. Neuron apoptosis, neural stem-cell (NSC) differentiation and myogenesis were also shown to be regulated by miR-1 [107,108]. miR-206 is associated with myogenic differentiation by targeting Pax7 and HDAC4 [108]. This miRNA not only plays a role in muscle development, but it also regulates repair of the neuromuscular junction and promotes reinnervation [109]. Additionally, it is known that miR-26a induces muscle differentiation, but its specific role in the central nervous system is not well studied. A study using rats with CP revealed that miR-26a can alleviate brain injury and inhibit apoptosis of brain cells and activation of astrocytes [106]. Li et al. and others have reported that miR-26 promotes axon growth through suppressing PTEN levels [110], further suggesting its potential role in CP.

Overexpression of miR-135b alleviated brain injury in hypoxia/ischemia-induced cerebral palsy rats by promoting NSC differentiation and inhibiting NSC apoptosis. miR135b, which is involved in myogenesis through regulating the IRS/PI3K pathway [111], was also found to be downregulated in CP [112].

In a rat model of CP, neurobehavioral effects and oxidative stress injury were alleviated and neural apoptosis was decreased through miR-200a overexpression, which limited Myt1L expression which is reduced in CP rats [113]. Furthermore, miR-200c overexpression inhibited skeletal muscle differentiation [114]. Satellite-cell proliferation is stimulated by the overexpression of miR-29c, which represses the expression of the atrophy-related genes, implying an association between lower miR-29 levels and CP [105,115]. This miRNA has been implicated in regulation of extracellular matrix deposition and its expression may be inhibited as a result of inflammation. Table 3 and Table 4 summarize miRNAs involved in neurogenic and myogenic processes.

CP patients have disrupted NMJ microanatomy [18,45]. Therefore, we identified the miRNAs associated with NMJ signaling and repair, synaptic growth, and synaptogenesis as listed in Table 5. Although a number of these candidates are yet to experimentally tested in the context of CP [109,139,140,141], their altered expression is expected to affect the onset and/or progression of CP due to their role in critically related processes.

### 4.2. Long Noncoding RNAs in CP

Long noncoding RNAs (lncRNAs) are transcripts longer than 200 nucleotides that control cellular processes at different levels [103]. Several lncRNAs have been shown to participate in the progression of an inflammatory response, endocrine disorders, and cancers [147]. Although a very small fraction of lncRNAs have been studied in CP, a number of lncRNAs are reported to be involved in the regulation of neural processes and myogenesis (see Table 6). For example, Li et al., used a neonatal rat model of hypoxic–ischemic (HI), to demonstrate that lncRNA-MIAT overexpression mitigates neuron apoptosis in HI-induced neonatal cerebral palsy by miR-211/GDNF [148]. Another lncRNA involved in neurodiseases is lncRNA hsrω, which participates in neurodevelopment at neuromuscular junctions by targeting FUS (Fused in sarcoma) protein [149]. Zhan et al. revealed that lncR-125b sponges miR-125b, which triggers differentiation of skeletal muscle satellite cells by increasing IGF2 expression in SCs [150]. Studies in Drosophila models demonstrated that inhibition of miR-125 negatively regulates NMJ function and phenotype, and the inhibition of miR-124 and let-7 causes the same defects in NMJs [142]. Furthermore, it has been found that there is an interaction between miR-1 and lncRNA-MALAT1. This lncRNA acts as a competitive endogenous RNA and neuroprotector regulating cell proliferation, migration, and inhibiting neuron apoptosis [107]. Moreover, lncRNA-MALAT1 also regulates proliferation and differentiation of satellite cells [151].

### 4.3. Circular RNAs in CP

Circular RNAs (circRNAs) are noncoding RNAs generated by a phenomenon called back splicing which leads to formation of covalently closed circles [167]. These circRNAs are the most recently appreciated class of regulatory RNAs primarily working by acting as sponges for miRNAs and RNA binding proteins [168]. These are highly enriched in the brain and involved in various neurological disorders [169]. Studies in patients with epilepsy and animal models of epilepsy have revealed altered ncRNA expression profiles [170], and recent studies have shown that circRNAs may also assist as biomarkers [171]. These ncRNAs could hold potential as biomarkers to predict CP, disease trajectory, or treatment responses [104,164].

CircRNAs have been shown to play a role in muscle development using animal models including chicken, bovine, and mouse; however, the field of circRNAs has not been fully explored yet regarding myogenesis and NMJ disruption. Table 7 summarizes circRNAs found in myogenesis in chicken (circSVIL, circFGFR2, circHIPK3, circRBFOX2s, circTMTC1), bovine (circLMO7, circFUT10, circFGFR4) and mouse (circZfp609). In addition, CDR1as is a well-studied circular RNA, highly expressed in the brain and enriched in skeletal muscle. CDR1as boosts insulin-like growth factor 1 receptor (IGF1R) levels via sponging miR-7, which triggers myogenic differentiation. CDR1 expression is induced by MyoD when satellite cells are transitioning from proliferation to differentiation [172]. Finally, although noncoding RNAs usually lack protein-coding ability, some of them can encode short micropeptides [103]. For example, circFAM188B is able to encode a peptide that promotes avian SC proliferation while repressing myogenesis [173].

## 5. Discussion

Our comprehensive review identified studies linking epigenetic phenomena with CP, including studies that employed CP research models and studies associated with muscle and movement control systems. We found reports linking epigenetics to CP, either directly or indirectly through effects of critical systems associated with CP, in all three major epigenetic systems that we looked at (i.e., DNA methylation, histone modification, and noncoding RNA). Within the identified reports, an emerging view is developing that epigenetic phenomena may either contribute to manifestations of CP or reflect the status of different tissues and cells including skeletal muscle. As clinicians and scientists seek to develop personalized medical approaches for CP, it will be important to identify the critical pathways involved in the onset and progression of disease.

It should be noted that at this early stage of epigenetics research in CP, detailed comparisons of results across different CP studies are challenging, and the identified literature illuminates some of the key challenges facing CP researchers. For example, CP is a spectrum of disease states, and specificity regarding the composition of the CP cohort(s) included in studies is critical to the output of the study and the specific target pathways identified. Similarly, details regarding the nature of the control cohort(s) used for comparisons may critically affect study outcomes. Achieving clinically relevant biomarkers (e.g., for diagnosis or for prognosis) or advanced therapies will likely require critically designed studies, and the analytic approach used to find, filter, and validate significant results could affect the direction of future research and development. In addition, in studies of muscle tissue and cells, questions regarding the degree to which factors such as participant demographics (e.g., age and sex), the manner and site of sample collection (e.g., surgical versus postmortem acquisition; anatomic sites), and the specific characterization of the tissues or isolated cells used as experimental models (e.g., fibrosis, fatty infiltration, and muscle cross-sectional area in tissue; positivity for Pax7, Myf5, NCAM1, and CXCR4 in cells) are not yet resolved. Thus, although most studies are well controlled internally, meta-analyses are difficult, and future efforts would greatly benefit from larger cohorts with common definitions for study design, analytic criteria, and reporting. Nonetheless, emerging data support the idea that epigenetics plays a key role in CP.

Although the field is new and rapidly developing, existing studies support the idea that DNA methylation patterns differ in peripheral blood cells of study participants with CP and suggest that such differences may extend to skeletal muscle tissues (Table 1). The results also suggest that aspects of these pattern differences may prove useful as biomarkers for identifying individuals with CP, especially when advanced computational approaches, such as machine learning, are employed [42,77]. In addition, a large and growing number of studies of the broader context in which DNA methylation differences arise suggest that DNA methylation differences in CP could be expected in cases arising due to perinatal inflammation, infection, or hypoxia and could even be associated with the onset or progression of disease. Finally, the prospect that altered DNA methylation may contribute to symptomology or pathology in CP is being investigated, and the possibility that DNA methylation pathways may offer therapeutic targets is compelling [74]. Significantly more research is needed to investigate these possibilities.

Little work has been accomplished to date in the analysis of histone modifications in CP (Table 2). There are strong but indirect indications that histone modifications may be critical in CP, including linkages between HDAC inhibitors and muscle regeneration [89,90] and the role of histone modifications in NMJs and in muscle satellite-cell function [91,92,93,94,95,96]. Further, there are indications that histone modifications are sensitive to conditions associated with CP such as hypoxia/ischemia [97], underscoring the need for additional investigation in this area. Histones play an important role in epigenetic regulation, and their modifications are related to gene activity state. Acetylation and phosphorylation of histones are generally correlated with transcriptional activation, while the effect of methylation depends on the level and residue [181]. Histone modifications can be studied in various ways, including mass spectrometry and ELISA for global changes, and chromatin immunoprecipitation (ChIP) for analysis of modifications on specific genome loci [181]. While studies of histone modifications in CP are limited, their critical role in muscle development and function, sensitivity to acute hypoxia and chronic ischemia, and importance as prognostic tools in other disease states support the evaluation of histone modifications to explore the epigenome state of individuals with CP.

Among the most compelling results of our review were the findings related to noncoding RNAs. Our searches identified a large set of miRNAs, some lncRNAs, and a few circRNAs that have been studied either directly for CP or have potentially associated roles due to their involvement in NMJ microanatomy and myogenesis. Future studies are needed to identify the detailed role of these and other ncRNAs in the genesis and progression of CP. Another critical unmet need is to understand the interplay between different classes of ncRNAs. Several circRNA:miRNA:mRNA or circRNA:lncRNA regulatory circuits have recently been discovered in a number of disorders, especially cancer [101,170], and similar studies are needed in CP. New findings that establish connections between ncRNAs, epigenetic regulation, and cell phenotype would help to understand mechanisms involved in diseases like CP that are challenging to study, and noncoding RNAs could provide advanced diagnostic and prognostic biomarkers to predict CP and its course or to serve as targets for new therapeutic approaches.

In numerous other disease states and conditions, an improved understanding of epigenetic involvement is leading to advanced diagnostics associated with epigenetic patterns and to new therapeutic approaches that seek to reset epigenetic alterations contributing to pathology. As epigenomic studies of CP continue to emerge, prospects for similar advances may be on the horizon, but significantly more work is needed to define relevant epigenetic phenomena, in order to understand how these phenomena arise and impact individuals across the complex spectrum of CP “phenotypes”, and to develop critical studies that identify clinically relevant biomarkers and that accurately define therapeutic targets that can be deployed to improve the health and care of individuals with different types of CP.

## Figures and Tables

**Table 1 jpm-11-01187-t001:** Studies associating DNA methylation with CP *.

Article PMID	Year	Title	Study Cohort(s)	Reference
34559924	2021	Differential DNA methylation and transcriptional signatures characterize impairment of muscle stem cells in pediatric human muscle contractures after brain injury	CP: 9.3 ± 4.5 years 5 ♂; 2 ♀ Non-CP: 14.5 ± 1.4 years 4 ♂; 4 ♀	[74]
33674671	2021	Whole genome methylation and transcriptome analyses to identify risk for cerebral palsy (CP) in extremely low gestational age neonates (ELGAN)	CP: newborns 23 ♂; 24 ♀ Non-CP: newborns 23 ♂; 24 ♀	[75]
32582584	2020	Epigenetic Marks at the Ribosomal DNA Promoter in Skeletal Muscle Are Negatively Associated with Degree of Impairment in Cerebral Palsy	CP: 9–18 years 16 ♂; 3 ♀ Non-CP: 7–21 years 8 ♂; 2 ♀	[76]
31035542	2019	Deep Learning/Artificial Intelligence and Blood-Based DNA Epigenomic Prediction of Cerebral Palsy	CP: newborns ** 15 ♂; 8 ♀ Non-CP: newborns ** 12 ♂; 9 ♀	[77]
30386170	2018	Preterm Birth and the Risk of Neurodevelopmental Disorders—Is There a Role for Epigenetic Dysregulation?	Review article	[73]
29925314	2018	Epigenetic machine learning: utilizing DNA methylation patterns to predict spastic cerebral palsy	CP: 14.7 ± 3.3 years 13 ♂; 3 ♀ Non-CP: 15.0 ± 2.2 years 15 ♂; 1 ♀	[42]
29694232	2018	Loss of myogenic potential and fusion capacity of muscle stem cells isolated from contractured muscle in children with cerebral palsy	CP: 8.9 ± 4.2 years 5 ♂; 3 ♀ Non-CP: 15.4 ± 1.3 years 4 ♂; 4 ♀	[56]
29484035	2018	Epigenome-wide analysis in newborn blood spots from monozygotic twins discordant for cerebral palsy reveals consistent regional differences in DNA methylation	All: Newborns 16 CP-discordant monozygotic twin pairs 20 ♂; 12 ♀	[78]
29043999	2017	Study of global DNA methylation in monozygotic twins with cerebral palsy	All: Newborns 2 monozygotic twin pairs	[79]
29039597	2017	Whole-genome scale identification of methylation markers specific for cerebral palsy in monozygotic discordant twins	All: 3.3 ± 0.5 years 4 CP-discordant monozygotic twin pairs 2 ♂; 6 ♀	[80]
27743978	2017	Influence of maternal obesity on the long-term health of offspring	Review article	[72]
25973051	2015	Kank1 re-expression induced by 5-Aza-2’-deoxycytidine suppresses nasopharyngeal carcinoma cell proliferation and promotes apoptosis	Cultured human cells used ***	[70]

* Results derived from a PubMed search (((“2010/01/01” [Date-Publication]: “2021/08/01” [Date-Publication])) AND ((cerebral palsy) AND (DNA methylation)). One article addressing folate metabolism and DNA synthesis was excluded from the table: PMID 22299647 [81]. ** The number of male and female participants was estimated based on data within the reference; there were 23 CP cases and 21 controls analyzed in the study. *** Article included due to the association between the *Kank1* gene and CP.

**Table 2 jpm-11-01187-t002:** Studies associating histone modification with CP *.

Article PMID	Year	Title	Study Cohort(s)	Reference
30386170	2018	Preterm Birth and the Risk of Neurodevelopmental Disorders—Is There a Role for Epigenetic Dysregulation?	Review article	[73]
25847581	2015	ELP2 is a novel gene implicated in neurodevelopmental disabilities	Two brothers with spastic diplegia **	[98]
24904523	2014	Insulin-Like Growth Factor Receptor Signaling is Necessary for Epidermal Growth Factor Mediated Proliferation of SVZ Neural Precursors in vitro Following Neonatal Hypoxia-Ischemia	Cultured rodent cells used	[99]

* Results derived from a PubMed search ((((“2010/01/01” [Date-Publication]: “2021/08/01” [Date-Publication])) AND ((cerebral palsy)) AND ((histone modification) OR (histone acetylation) OR (histone methylation) OR (histone phosphorylation) OR (histone ubiquitylation) OR (histone deacetylase)))). ** Article included due to the presence of CP symptoms (spastic diplegia) in the study subjects and the association between the ELP2 gene and histone acetylation.

**Table 3 jpm-11-01187-t003:** MicroRNAs in neural stem cells and neurogenesis *.

microRNA	Role	Target(s)	Reference
miR-1	Promotes neuron apoptosis & NSC differentiation	Hsp70s, Hes1	[107]
miR-7	Neurogenesis	NLRP3/caspase-1	[106]
miR-19	Promotes NSC proliferation	FoxO1	[106]
miR-26a	Promotes neurite outgrowth	PTEN	[105]
miR-128	Promotes neural differentiation	UPF1, MLN51	[116]
miR-135b	Promotes NSCs differentiation	S100B	[112]
miR-200	Decreases neuron apoptosis	Mytl1	[113]

* Results derived from a PubMed search (((“2010/01/01” [Date-Publication]: “2021/08/01” [Date-Publication])) AND ((cerebral palsy) AND (miR)). Articles addressing other noncoding RNAs and other diseases (neurological complications such as inflammation, posthemorrhagic hydrocephalus) were excluded.

**Table 4 jpm-11-01187-t004:** MicroRNAs in myogenesis *.

microRNA	Role	Target(s)	Reference
miR-27a/b	Regulator of cell quiescence	SCs activation (Mstn)	[117]
miR-378	Regulator of cell quiescence	Delay SCs activation (Igf1r)	[118]
miR-489	Regulator of cell quiescence	Maintain SC quiescence (Dek)	[119]
miR-708	Regulator of cell quiescence	Maintain SC quiescence (Tns3)	[120]
miR-27a/b	Promoter of cell proliferation	Mstn	[117,121]
miR-29c	Promoter of cell proliferation	MuRF1, Atrogin-1, HDAC4	[115]
miR-99a-5p	Promoter of cell proliferation	MTMR3	[122]
miR-133	Promoter of cell proliferation	SRF	[123]
miR-192	Promoter of cell proliferation	RB1	[124]
miR-221	Promoter of cell proliferation	P27, P57	[123]
miR-2400	Promoter of cell proliferation	MYOG	[125]
miR-2425-5p	Promoter of cell proliferation	RAD9A, MYOG	[126]
miR-1/206	Inhibitor of cell proliferation	HDAC4, Pax7	[108]
miR-9-5p	Inhibitor of cell proliferation	IGF2BP3	[127]
miR-27b	Inhibitor of cell proliferation	MDFI	[128]
miR-34c	Inhibitor of cell proliferation	Notch1	[129]
miR-128	Inhibitor of cell proliferation	Sp1	[130]
miR-143	Inhibitor of cell proliferation	IGFBP5	[131]
miR-199b	Inhibitor of cell proliferation	JAG1	[132]
miR-1/206	Promoter of myogenic differentiation	HDAC4, Pax7	[108,133]
miR-17/19	Promoter of myogenic differentiation	Ccnd2, Jak1, Rhoc	[134]
miR-26a	Promoter of myogenic differentiation	TGFb, BMP	[106]
miR-27b	Promoter of myogenic differentiation	MDFI, Pax3	[128]
miR-34c	Promoter of myogenic differentiation	Notch1	[129]
miR-92	Promoter of myogenic differentiation	DKK3	[135]
miR-127	Promoter of myogenic differentiation	S1PR3	[136]
miR-133	Promoter of myogenic differentiation	UCP2	[123]
miR-139	Promoter of myogenic differentiation	DHFR	[137]
miR-486	Promoter of myogenic differentiation	Pax7	[138]

* Results derived from a PubMed search (((“2010/01/01” [Date-Publication]: “2021/08/01” [Date-Publication])) AND ((muscle satellite cells) AND (miR)). Articles that had no clear and related results were excluded. Articles that were written in other languages were excluded as well.

**Table 5 jpm-11-01187-t005:** MicroRNAs in Neuromuscular Junction *.

microRNA	Role	Reference
let-7	NMJ function and phenotype (Drosophila) **	[142]
miR-2	NMJ signaling (mouse) **	[143]
miR-8	Synaptic growth at the NMJ (Drosophila) **	[142]
miR-23a	Neuroprotection and prevention of muscle-fiber atrophy (mouse) **	[144]
miR-34	Synaptogenesis (Drosophila) **	[145]
miR-124	NMJ function and phenotype (Drosophila) **	[142]
miR-125	NMJ function and phenotype (Drosophila) **	[142]
miR-126-5p	Rescue of axon degeneration and NMJ disruption (mouse) **	[139]
miR-153	Synaptic homeostasis (Drosophila) **	[142]
miR-206	Maintenance and repair of NMJ (mouse) **	[109]
miR-219	Abnormalities at the NMJ (mouse) **	[146]
miR-234	Resistance to acetylcholinesterase inhibitor aldicarb/neuropeptide release (C. elegans) **	[140]
miR-289	Synaptic growth at the NMJ (Drosophila) **	[142]
miR-310/313	Neurotransmitter release (Drosophila)	[141]
miR-958	Synaptic growth at the NMJ (Drosophila) **	[142]

* Results derived from a PubMed search ((((“2010/01/01” [Date-Publication]: “2021/08/01” [Date-Publication])) AND ((neuromuscular junction) AND (miR)). Articles not involving miRNAs and articles about other diseases such as Myasthenia gravis, Charcot-Marie-Tooth disease, among others were excluded. ** Type of nonhuman model system used is given in parentheses.

**Table 6 jpm-11-01187-t006:** Long noncoding RNA in myogenesis *.

lncRNA	Role	Target(s)	Reference
lncRNA MyHC IIA/X-AS	Promotes SC differentiation	miR-130b	[152]
lnc23	Promotes SC differentiation	PFN1	[153]
lncR-125b	Promotes SC differentiation	miR-125b	[150]
lnc-H19	Promotes SC differentiation	miR-140-5p, Sirt1/FoxO1, TDP43	[154,155,156]
lncR-MSTRG.59589	Promotes SC differentiation	PALLD	[157]
linc-RAM	Promotes SC differentiation	MyoD	[158]
lncMyoD	Promotes SC differentiation	MyoD	[159]
lncRNA-MUNC	Promotes SC differentiation	MyoD	[160]
linc-YY1	Promotes SC differentiation	YY1	[161]
lnc403	Inhibits SC differentiation	Myf6	[162]
lncR-SAM	Promotes myoblast proliferation	Sugt1	[163]
lnc133b	Promotes SC proliferation	miR-133b	[164]
lncRNA-MALAT	Promotes SC proliferation	miR-1	[107]
CTTN-IT1	Promotes SC proliferation	miR-29a	[165]
linc-YY1	SC activation/proliferation	Pax7	[166]

* Results derived from a PubMed search (“2010/01/01” [Date-Publication]: “2021/08/01” [Date-Publication]) AND ((muscle satellite cells) AND (lncRNA)). Repeated articles and articles that were not focused on related concepts were excluded.

**Table 7 jpm-11-01187-t007:** Circular RNAs in myogenesis *.

circRNA	Role	Target(s)	Reference
CDR1as	Promotes SCs differentiation	miR-7	[172]
circSVIL	Promotes SCs differentiation	miR-203	[174]
circFGFR2	Promotes muscle proliferation and differentiation	miR-133a-5p, miR-29b-1-5p	[151]
circHIPK3	Promotes proliferation and differentiation of myoblasts	miR-30a-3p	[175]
circRBFOX2s	Promotes proliferation of myoblasts	miR-1a-3p, miR-206	[176]
circTMTC1	Inhibits SC differentiation	miR-128-3p	[177]
circLMO7	Inhibits SC differentiation/Promotes cell proliferation	miR-378-3p	[178]
circFUT10	Inhibits SC proliferation/Promotes SC differentiation	miR-133a	[179]
circFGFR4	Promotes SC differentiation	miR-107	[180]
circZfp609	Inhibits myoblast differentiation	miR-194-5p	[146]

* Results derived from a PubMed search ((“2010/01/01” [Date-Publication]: “2021/08/01” [Date-Publication])) AND ((muscle satellite cells) AND (circRNA)). Additionally, results derived from a PubMed search (((“2010/01/01” [Date-Publication]: “2021/08/01”[Date-Publication])) AND ((muscle satellite cells) AND (miR)) were included as well; these articles were focused on the role of circRNAs in regulating the miRNAs.

## Data Availability

Not applicable.

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
