# Peer review of "An Emerging Role for Epigenetics in Cerebral Palsy"

_jpm, 2021, doi:10.3390/jpm11111187_

Round 1
Reviewer 1 Report
The goal of the paper titled "An emerging role for epigenetics in cerebral palsy" is to provide a review of the growing literature indicating that epigenetic phenomena are increasingly being considered as playing roles particularly in the phenotypic presentations of cerebral Palsy (CP). The paper adds to the body of literature on CP because it summarizes the first several phenomena which have been associated with CP, such as limited MRI findings in children ultimately diagnosed with CP, and changes in muscle architecture in most children diagnosed with CP. Then the article outlines and reviews multiple papers that have tried to disentangle the knowledge surrounding these phenomena by characterizing the potential for epigenetic impacts, such as histone modification s part of gene expression and differential patterns of DNA methylation in children with CP.
The paper is limited in my opinion, by 3 issues:
- overestimating the prevalence of CP. The authors state that CP occurs in 1 in every 500 births. The authors do provide a citation, but most prevalence rates are lower. The authors perhaps need to examine more literature and provide a range.
- clearly defining epigenetics versus genetic impacts. The introduction would be strengthened, by providing the differential definitions of why understanding epigenetics is important in CP. The target audience for this paper will primarily be clinicians many of which will not have a clear understanding of this distinction. Most of the field is currently focused on understanding if there are in fact genetic causative factors associated with CP. The authors acknowledge this, but what they fail to explain is that the reason epigenetics is important is that with almost all 'disease' (for lack of a better term) states, genetics sets up the potential and epigenetics plays the ultimate causal role in gene expression. This is what we are finding for CP, and in fact, the review is needed because the many variations of CP is being linked to a myriad of phenotypic expressions. Adding, the reasoning in clearer detail in the introduction would help readers.
- The article would also benefit from a more clearly outlining what we do not know and a future directions section for example, because CP is caused very early in life, epigenetics and 'personalized medicine is equally (maybe more so) for therapeutics and diagnostic. In my opinion, epigenetics has focused on understanding causes, but CP is a 'static' encephalophy. The power of epigenetics is in disentangling the cellular response and this needs to be more tied to the therapeutic side of the equation.
In short, the paper is well-written but could be enhanced by better considering a potential reader versus just listing the papers involved in the review.
This is a well-presented article that adds to the overall discussion about the potential causes of CP. My only concern is that the author's prevalence rating for CP is what would be considered high by most in the field. The primary citation they provided for prevalence is for the co-occurrence of autism and CP. The authors need to expand their prevalence estimates to include other citations, at which point. I believe their estimations will go up.
Reviewer 2 Report
A wonderfully written research article. Very contemporary and on the pulse with the discussions emerging in CP. Well thought out, researched, and written.
The paper has explored the studies written on the epigenetic changes seen in children with CP. They have outlined several epigenetic pathways that have been shown to be affected by hypoxia and inflammation in children with CP. These are pathways that could potentially be intervened with in the future.
The main question is ' do epigenetics play a role in CP - and this is very relevant to discussions that are on the CP agenda with regards to etiology and the interaction between environmental risk factors and genetic risk factors. It is certainly very relevant and interesting.
It is original in that it summarises the current literature - I am not aware that another such summary has recently been published.
The paper is well written and easy to follow. The article is well referenced with current literature.
The conclusions are evidenced-based and whilst the CP scientific community is split - those that believe genetics play a role in all kids vs those who believe it plays a role in only a select - this gives compelling evidence that the interplay between environment and genetics is far more complicated than is currently understood.
I am happy with the article as is.
Author Response
We thank the reviewer for their kind and supportive comments. Some changes were made in the revision that we believe strengthen the manuscript.